# Screening of Poly-Glutamic Acid (PGA)-Producing *Bacillus* Species from Indian Fermented Soybean Foods and Characterization of PGA

Priyambada Pariyar [1], Puneeta Singh Yaduvanshi [2], Pullakhanadam Raghu [2] and Jyoti Prakash Tamang [1,*]

1   Department of Microbiology, School of Life Sciences, Sikkim University, Gangtok 737102, India
2   Micronutrient Research Division, ICMR-National Institute of Nutrition, Osmania, Hyderabad 500007, India
*   Correspondence: jptamang@cus.ac.in; Tel.: +91-9832061073

**Abstract:** This present work is aimed to screen the PGA-producing *Bacillus* spp. from naturally fermented soybean foods of Arunachal Pradesh in India and to characterize the poly-glutamic acid (PGA) extracted from *Bacillus* spp. A total of 50 isolates were screened for high stickiness from 165 bacterial isolates isolated from 34 samples *viz.*, *grep chhurpi*, *peha/paeha*, *peron namsing*, and *peruñyaan*. Based on 16S rRNA gene sequence analysis, 50 isolates were identified as *Bacillus* spp. Viscosity of 50 species of *Bacillus* were measured, out of which 7 species of *Bacillus* showing ≥0.03 Pa.s (30 cP) of viscosity were selected *viz.*, *Bacillus velezensis* GC1-42, *B. siamensis* GC4-36, *B. subtilis* PH3-21, *B. subtilis* PN1-14, *B. subtilis* PH6-29, *B. tequilensis* PN9-22, and *B. safensis* subsp. *safensis* PY1-19 for PGA production. Amino acid analysis of PGA extracted from seven species of *Bacillus* showed high molecular weight (>600 kDa). *B. safensis* subsp. *safensis* PY1-19, isolated from *peruñyaan*, showed 78.9% of glutamic acid, antagonistic properties against pathogenic bacteria and had the ability to produce phytase with no hemolytic activity. Hence, this strain was selected as a potential starter to prepare *peruñyaan* at laboratory, and the final product showed high stickiness and viscosity with production of PGA of around 11 g/L.

**Keywords:** fermented soybean; polyglutamic acid; *Bacillus* spp.; *B. safensis* subsp. *safensis*; starter culture

## 1. Introduction

Fermentation is an important human discovery that has enabled food preservation and flavor enhancement, along with additional functional properties imparted by the organisms, which in turn benefit consumers [1]. Traditionally soybeans are fermented into different food products in Asia, which are mainly fermented by domesticated molds [2] or bacteria, mostly *Bacillus* spp. [3]. Fermented soybean foods, mostly dominated by *Bacillus* spp., are consumed as a delicacy in South East Asia [4]. However, consumption of bacterial-fermented soybean foods with mucilaginous texture and *umami* [5] flavor are traditional dietary items of the ethnic people belonging to the Mongolian races in the North East states of India, whereas such fermented soybean foods are traditionally and historically not consumed in other parts of India [6]. One of the featured characteristics of bacterial-fermented soybean foods is the appearance of mucilaginous, or sticky materials, on the surface of fermented soybeans, which is the preferred quality criterion for the final product by the consumers [7]. More stickiness indicates the good quality of the product. Mucilaginous or sticky materials appearing during soybean fermentation on the surface of soybeans is actually poly-γ-glutamic acid (γ-PGA) produced by *Bacillus* spp. [8,9] in many Asian fermented soybean foods, such as *kinema*, *hawaijar*, *bekang* and *tungrymbai* of India, *natto* of Japan, *pe poke* of Myanmar, and *cheonggukjang* of Korea [10,11]. The poly-glutamic acid (PGA) is a polymer that is composed of D- and L-glutamic acid units linked via γ-amide linkages and is produced exogenously by *Bacillus* spp. [12]. Polyglutamic acid has several bio-functional properties [13,14] and health benefits for consumers [15]. Recent

studies on PGA-rich fermented soybean foods have shown an ability of PGA to suppress increased postprandial blood glucose levels [16]. PGA is capable of regulating blood pressure due to its ability to inhibit angiotensin-converting enzyme (ACE) activity [17] and is associated with intestinal calcium absorption as well [18]. It is also a biological glue that has major applications in wound healing and treatment of dental caries [12].

Naturally fermented soybean foods, which are alkaline in nature, are important low-cost digested protein foods in the gastronomy of ethnic people of Arunachal Pradesh, one of the states of North East India located in the Eastern Himalayas (Figure 1). Some common naturally fermented soybean foods of Arunachal Pradesh are *grep chhurpi*, *peha/paeha*, *peron namsing*, and *peruñyaan*, which are prepared by different ethnic communities using their ethno-microbiological knowledge of food fermentation and consumed as curry or soup. The traditional methods of preparation of these ethnic fermented soybean foods are quite similar with each other; however, there are some variations, such as their use of wrapping materials, fermentation periods, and post-fermentation storage (Figure 1). *Grep chhurpi* and *peha/paeha* are further crushed and dried after fermentation and rolled into a ball before they are consumed. Dried *grep chhurpi* and *peha/paeha* can be stored at room temperature for 6-9 months, whereas *peron namsing* and *peruñyaan* are consumed immediately after the fermentation due to minimum shelf-life of four days. Sometimes, freshly fermented *peron namsing* and *peruñyaan* are sun-dried and can be stored at room temperature for 6-9 months (Figure 2).

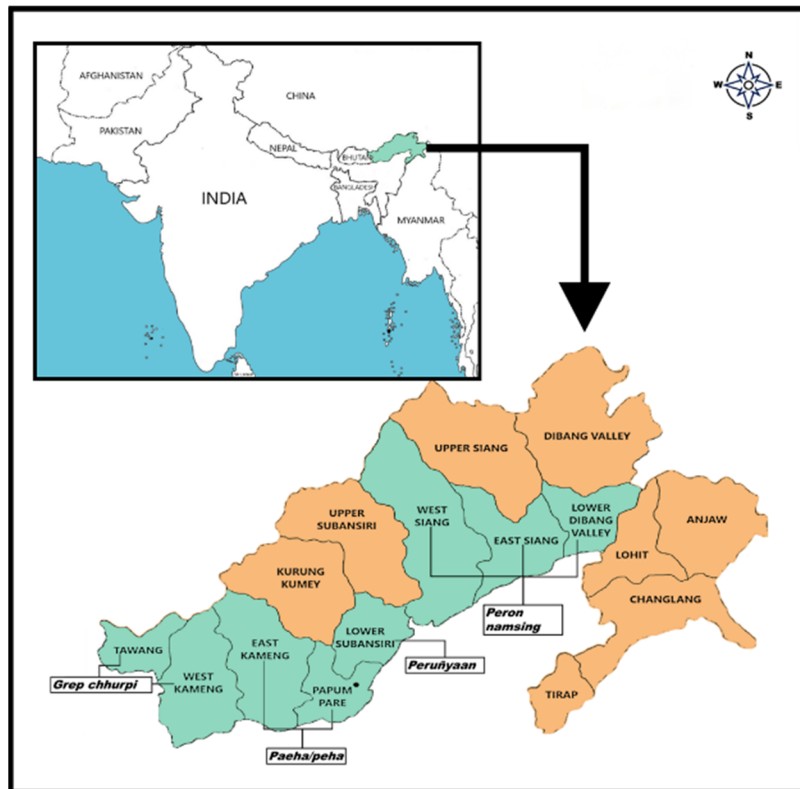

**Figure 1.** Sample collection sites in Arunachal Pradesh, India.

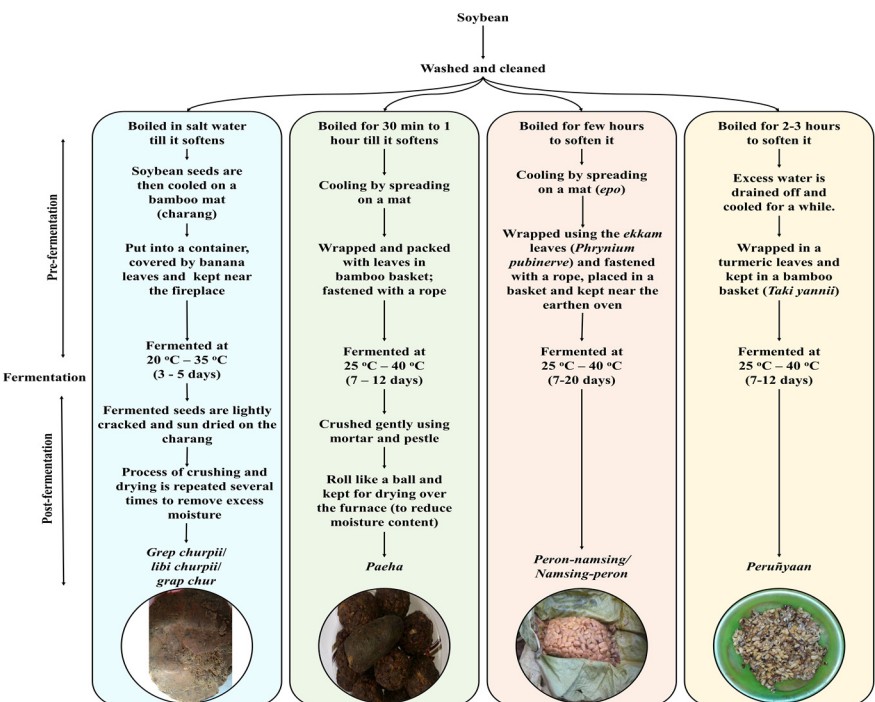

**Figure 2.** Traditional methods of preparation of the naturally fermented soybean products of Arunachal Pradesh, India: *grep chhurpi, peha/paeha, peron naming,* and *peruñyaan.*

There is no report on microbial composition of rare and unique fermented soybean foods of Arunachal Pradesh. Since it has been proved that PGA-producing *Bacillus* is the dominant bacterium in naturally fermented soybean foods [19]; hence, this present work is aimed to screen and identify the dominant PGA-producing *Bacillus* spp. from naturally fermented alkaline soybean foods (*grep chhurpi*, *peha/paeha*, *peron namsing* and *peruñyaan*) of Arunachal Pradesh in India. It is also aimed to determine molecular weight and glutamic acid content of PGA extracted from *Bacillus* spp. and to select the high PGA-producing *Bacillus* sp. for development of a starter culture.

## 2. Materials and Methods

### 2.1. Collection of Samples

Thirty-four dried samples of naturally fermented soybean foods were collected from various local markets of Arunachal Pradesh viz., *grep chhurpi* (Tawang), *peha/paeha* (Papum Pare), *peron namsing* (East Siang), and *peruñyaan* (Lower Subansiri) (Figure 1). Samples were collected aseptically in pre-sterilized containers and ferried to the laboratory in an ice-box cooler and stored at 4 °C for further analysis.

### 2.2. Analysis of pH

One gram of sample was dissolved in 10 mL sterile distilled water, and the pH was measured using a pH meter (GeNei™, Bangalore, India) and calibrated with standard buffers. The pH value was represented as mean ± SD values of triplicates sets.

### 2.3. Isolation of Bacteria

Ten grams of each sample were homogenized in 90 mL of sterilized physiological saline using the stomacher lab-blender (400, Seward, UK) for 1 min. Serial dilutions were made and heated at 100 °C for 2 min to inactivate vegetative cells of endospore bacteria [8]. Dilutions were enumerated on nutrient agar (NA) (M001, HiMedia, India) and incubated for 24 h at 37 °C. Colonies that appeared in NA plates were selected randomly. Number of colonies was counted as colony forming unit (CFU). Purity of the isolates was checked by

streaking again on fresh NA plates followed by microscopic examinations and then stored at 20% glycerol in − 20 °C for further analysis.

### 2.4. Measurement of Stickiness

Overnight cultures of isolates in phytone agar incubated at 37 °C were used for determining the ability to produce stickiness. Sticky thread was pulled from each isolate by a sterile inoculating needle and the length of the thread was measured in cm [8] in triplicates.

### 2.5. Measurement of Viscosity

Bacterial isolates showing high stickiness (>20 cm) were selected and were grown in PGA isolation broth [7] consisting of 100 mL of each broth: 5 g of L-glutamic acid (GRM048, HiMedia, India), 5 g of glucose (MB037, HiMedia, India), 0.1 g of $Na_2HPO_4.12H_2O$ (Merck, India), 0.1 g of $MgSO_4.7H_2O$ (Merck, India), 0.1 g of $KH_2PO_4$ (Merck, India), 0.002 g of $MnSO_4.H_2O$ (Merck, India), 0.005 g of $FeCl_3.6H_2O$ (Merck, India), 0.02 g of $CaCl_2$ (Merck, India) and 50 μg of biotin (CMS095, HiMedia, India). The cultures were then incubated at 37 °C for 3 days at 120 rpm and 30 mL of the aliquot was taken and the dynamic viscosity was measured at using a viscometer (DV1MRVTJ0, Brookfield AMETEK, MA, USA) in triplicates. Further, methanol was added to the cell free filtrate in a 4:1 ratio (*v/v*) and the white fibrous precipitate was collected by centrifugation at 5000 rpm for 20 min. After the supernatant is discarded, the collected pellet was then incubated at 4 °C overnight [20]. This further confirmed their ability to produce PGA from the media.

### 2.6. Phenotypic Characteristics

Cell morphology and motility of the isolates were observed using a phase contrast microscope (Olympus, CKX41, Japan). Gram staining and catalase production from the isolates were determined [8].

### 2.7. Genotypic Identification

2.7.1. Genomic DNA Isolation

Genomic DNA was extracted using the method described by Shangpliang and Tamang (2021) [21] with slight modifications. Cultures were grown in broth at 37 °C for 16–18 h. Two mL of the inoculated broth was then transferred to a fresh microcentrifuge tube followed by centrifugation at 3000 rpm for 5 min. The pellet was washed twice with 0.5 M NaCl followed by deionized water (MilliQ $H_2O$) and then dissolved in 200 μL 1 × TE (Tris-EDTA) buffer. The cells were then subjected to enzymatic lysis using 10 μL of lysozyme (20 mg/mL) and incubated at 37 °C for 30 min, followed by heating at 95 °C for 20 min. The cell lysate was then collected by centrifugation at 4000 rpm for 10 min at 4 °C and transferred to a fresh sterile microcentrifuge tube. DNA quality was checked using Eppendorf Bio-Spectrometer (Model 6135 000 009, Hamburg, Germany) and absorbance at $A_{260}/A_{280}$ ratio of 1.8–2.2 was used for PCR amplification.

2.7.2. PCR Amplification

The PCR reaction was performed in a 50 uL reaction volume containing GoTaq® Green Master Mix (M7122, Promega, WI USA), Taq DNA polymerase, dNTPs, MgCl2, and reaction buffers. Amplification of the 16S rRNA gene was achieved using the universal primers (27F 5′- AGAGTTTGATCATGGCTCAG-3′ and 1492R 5′-GTTACCTTGTTACGACTT-3′) [22] in SimpliAmp™ Thermal Cycler (Cat No. A24811, ThermoFisher Scientific, Carlsbad, CA, USA). The PCR conditions were run as follows: initial denaturation for 5 min at 94 °C, followed by denaturation in 30 cycles of 45 s at 94 °C, annealing for 45 s at 55 °C, then the elongation process for 1.5 min at 72 °C, and final elongation at 72 °C for 10 min, with a stoppage reaction at 4 °C [21]. Amplification of the target 16S rRNA gene was confirmed using 1.4% agarose and visualized using Gel Doc™ EZ Imager (BioRad, Hercules, CA, USA).

### 2.7.3. Purification of PCR Amplicons

Purification of the PCR amplicons was performed using the slightly modified method described by Shangpliang and Tamang [21], using PEG (polyethylene glycol)-NaCl and 20% (*w/v*) PEG, 2.5 M NaCl. The PCR amplicons were mixed with 0.6 volumes of PEG-NaCl and incubated for 30 min at 37 °C. The mixture was then centrifuged at 4 °C for 30 min at 12,000 rpm. The aqueous supernatant was discarded carefully without disturbing the pellet, followed by washing with 70% ethanol and air-dried at room temperature. Finally, 20 μL of nuclease free water was added to purified DNA and quality of the DNA was checked in 1.4% agarose gel and visualized using Gel Doc™ EZ Imager (BioRad, Hercules, CA, USA).

### 2.7.4. 16S rRNA Gene Sequencing

Sequencing of the purified PCR products was achieved using an automated DNA analyzer (ABI 3730XL Capillary Sequencers, Applied Biosystems, Foster City, CA, USA). Here, two PCR reactions were performed using primers: 27F 5'-AGAGTTTGATCATGGCTCAG-3' and 1492R 5'-GTTACCTTGTTACGACTT-3' [22]. Each reaction set consisted of 0.2 μM primer, 0.2 mM dNTPs, 2.0 mM MgCl2, 0.5 mg/mL1, and 0.04 U/μL 1 Taq DNA polymerase for a final volume of 50 μL and PCR conditions of an initial denaturation of 10 min at 95 °C, with 35 cycles of denaturation of 1 min at 95 °C, followed by annealing of 2 min at 40 °C, and elongation at 72 °C for 1 min. The reaction was extended with a final elongation step of 10 min at 72 °C and stopped at 4 °C.

### *2.8. Characterization of PGA*

Bacterial isolates showing high viscosity of >0.03 Pa.s (30cP), as a threshold, were selected for the characterization of PGA. A purified commercial standard poly-L-glutamic acid (P4886, Sigma-Aldrich, MO, USA) (PGA) was used as a reference.

### 2.8.1. Extraction of PGA from Bacterial Isolates

Extraction of PGA from bacterial isolates was performed following the methods of l and group [23]. The 5 mL of each isolate was inoculated in the selective media (5 g of L-glutamic acid (HiMedia, India), 5 g of glucose (HiMedia, India), 0.1 g of $Na_2HPO_4.12H_2O$ (Merck, USA), 0.1 g of $MgSO_4.7H_2O$ (Merck, India), 0.1 g of $KH_2PO_4$ (Merck, USA), 0.002 g of $MnSO_4.H_2O$ (Merck, India), 0.005 g of $FeCl_3.6H_2O$ (Merck, USA), 0.02 g of $CaCl_2$ (Merck, USA), and 50 μg of biotin (HiMedia, India) and incubated at 37 °C for 72 h. The fermented media were then subjected to centrifugation at 20,000 rpm for 30 min at 4 °C to separate the biomass and unutilized materials. The supernatant was then mixed with four volumes of cold methanol and was again centrifuged at 20,000 rpm for 30 min at 4 °C. The precipitate was collected and subjected to rotary evaporation for the solvent to dry at 30 °C. The samples were then flushed with nitrogen and stored at − 20 °C, until further analysis.

### 2.8.2. Amino Acid Analysis

Amino acid analysis was carried out as described by Longvah et al. [24]. Ten mg protein was digested in 6N HCl overnight, excess acid was removed by continuous flash evaporation under reduced pressure (Buchi, Switzerland), and the sample was then dissolved in citrate buffer (pH 2.2). An aliquot (20 μL) of the sample was loaded into the automated amino acid analyzer (Biochrom-30, Cambridge, UK). Each amino acid was identified and quantified using authentic against standard (National Institute of Standards and Technology, SRM 2389).

### 2.8.3. Gel Filtration Analysis

The methanolic precipitate of PGA was dissolved in water, dialyzed against 10 mmol/L phosphate buffered saline pH 7.5 (PBS), and subjected to gel filtration chromatography on a TSK-G4000 $SW_{XL}$ (Sigma Chemical Co, Bangalore, India), column connected to HPLC (Model 1100, Agilent, USA), equilibrated, and eluted with PBS. The 10 μg of protein was subjected to size fractionation at a flow rate of 1 mL/min, and the elution was

monitored at 210 nm for protein. Thyroglobulin (660kDa, Amersham Biosceinces, UK), was also run under identical conditions as standard molecular weight.

### 2.9. Preparation of Peruñyaan Using Bacillus safensis Subsp. safensis PY1-19

*Bacillus safensis* subsp. *safensis* PY1-19, isolated from naturally fermented *peruñyaan*, which showed the highest amino acid content, was selected as a starter to prepare *peruñyaan* at laboratory (Figure 3). We prepared *peruñyaan* at laboratory condition following the methods described by Tamang in 1999 [25] for preparation of *kinema*, similar fermented soybean food of Sikkim. Seeds of yellow-colored soybeans (*Glycine max*) were cleaned and soaked overnight in ambient temperature and boiled at 121 °C for around an hour. After boiling, the excess water was drained off and the beans were allowed to cool down. Overnight grown *Bacillus safensis* subsp. *safensis* PY1-19 in nutrient broth, with ~$10^7$ CFU/g, was inoculated to the sterilized soybeans in pre-sterile beaker covered with thinly perforated poly-sheets and incubated at 40 °C for 72 h. The stickiness and viscosity of the final product were measured as mentioned Sections 2.4 and 2.5.

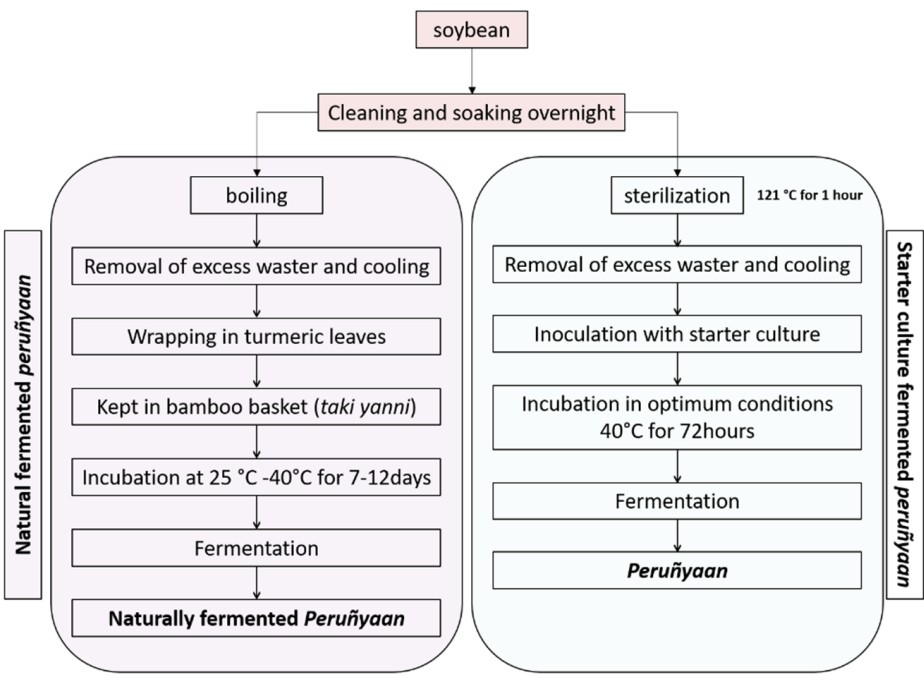

**Figure 3.** Comparative flowchart for the naturally fermented soybean food *peruñyaan* and starter culture (*B. safensis* subsp. *safensis* PY1-19) fermented-*peruñyaan*.

PGA Production

The functional groups in the prepared PGA were determined using Fourier-transform infrared (FTIR) spectroscopy reads at 4000–400 cm$^{-1}$ by identifying the values of the peaks in the graphical results obtained [26].

### 2.10. Characterization of B. safensis Subsp. safensis PY1-19

2.10.1. Hemolysis

Hemolytic reaction of *B. safensis* subsp. *safensis* was evaluated following the method described by Lee and Chang [27]. The bacterial culture was streaked on 5% blood agar plates and incubated at 37 °C overnight. The plates were examined for hemolysis.

2.10.2. Phytase Activity

Phytase activity was determined using the method as per Verma and group [28] with slight modification. *B. safensis* subsp. *safensis* was spotted on phytase screening media (Glucose 15 g/L (HiMedia, India), NH$_4$NO$_3$ 5.0 g/L (Merck, India), KCl 0.5 g/L

(Merck, India), $MgSO_4 \cdot 7H_2O$ 0.5 g/L (Merck, India), $FeSO_4 \cdot 7H_2O$ 10 mg/L (Merck, India), $MnSO_4 \cdot 7H_2O$ 10 mg/L (Merck, India), Na-phytate 0.05% (HiMedia, India), agar 20 g/L (HiMedia, India)) for 72 h at 30 °C. The diameter of the clear halo surrounding the colonies was measured.

### 2.10.3. Antagonistic Properties against Pathogenic Bacteria

The antimicrobial property of *B. safensis* subsp. *safensis* was determined using the agar well diffusion method [29]. *Bacillus cereus* MTCC 1272, *Escherichia coli* MCC 2413, *Salmonella enteric* subsp. *enterica* MTCC 3223, and *Staphylococcus aureus* MTCC 740 were used as the indicator organisms that were plated on Mueller Hinton Agar (M173, HiMedia, India) overlaying soft agar. About 100 µL of *B. safensis* subsp. *safensis* was poured into the wells prepared using a cork borer. The plates were then incubated overnight at 37 °C.

### 2.11. Bioinformatics Analysis

Analysis of the raw reads was performed as described by Shangpliang and Tamang, 2021 [24] with few modifications. Briefly, the quality of the raw sequences was initially checked using Sequence scanner v2.0 (Applied Biosystems). ChromasPro v1.34 (Technelysium Ltd., South Brisbane, Australia) was used to assemble the good quality sequences, whereas chimera sequences were checked using Mallard (http://www.download32.com/mallard-software.html). Furthermore, sequences sequence similarity was checked by aligning with BLAST (basic local alignment search tool) and EzTaxon databases to identify the isolates. ClustalW was used to align the sequences for identifying the phylogenetic relationship of the identified species [30], and phylogenetic tree was constructed using maximum-likelihood method based on the Kimura 2-parameter model [31] by Molecular Evolutionary Genetics Analysis version 7 (MEGA7.0.26) [32].

### 2.12. Statistics and Visualization

Significant differences in the log CFU/g and pH among the products were checked using paired Student's T-test. Furthermore, significant differences between the observed and the expected species frequencies were tested using Chi-squared test in MS-Excel v365. Additionally, to visualize the shared and unique *Bacillus* species identified, iGraph R-package was used. Alpha diversity was calculated using PAST v4.0 [33]. For visualizing the measurements of stickiness and dynamic viscosity tested, bar-chart and principal component analysis (PCA) were plotted using PAST v4.0.

## 3. Results

### 3.1. Bacillus Species Diversity

A total of 34 samples of naturally fermented soybean foods (4 samples of *grep chhurpi*, 10 samples of *peha/paeha*, 14 samples of *peron namsing* and 6 samples *peruñyaan* samples, respectively) were collected from various parts of Arunachal Pradesh (Figure 1) and their traditional methods of preparation and mode of consumption were documented (Figure 2). Naturally fermented soybean foods of Arunachal Pradesh are alkaline in nature with pH of *grep chhurpi* ranging from $7.4 \pm 0.28$ to $7.7 \pm 0.14$, *peha* $5.66 \pm 0.04$ to $8.4 \pm 0.01$, *peron namsing* $7.7 \pm 0.005$ to $8.6 \pm 0.005$, and *peruñyaan* $7.2 \pm 0.05$ to $8.2 \pm 0.1$ (data not shown). Bacterial populations in all samples were $7.9 \pm 0.15$ CFU/g to $8.2 \pm 0.22$ CFU/g (data not shown). A total of 165 bacterial isolates were isolated from 34 samples, out of which 50 isolates were screened for high stickiness of $\geq 20$ cm (Figure 4a). Based on the limited morphological and phenotypic tests, 50 endospore-forming bacteria were tentatively identified as *Bacillus*. The 16S rRNA gene sequencing analysis confirmed the identify of *Bacillus* spp. using the NCBI database and EzTaxon (Figure 5a). *Bacillus subtilis* was the predominant species in fermented soybean foods of Arunachal Pradesh, followed by *B. siamensis*, *B. velezensis*, *B. licheniformis*, *B. tequilensis*, *B. altitudinis*, *B. safensis* subsp. *safensis*, *B. paralicheniformis* (Figure 5b,c). *B. subtilis* was shared among *peron namsing*, *peha*, and *peruñyaan* but was not isolated from *grep chhurpi* (Figure 5b,c). On the contrary, *B. siamensis* was only isolated from

*grep chhurpi* and *peruñyaan*. *B. velezensis* and *B. licheniformis* were isolated from *peha*, *grep chhurpi*, and *peruñyaan* but were absent in *peron namsing*. *B. tequilensis* was isolated only from *peron namsing* and *peruñyaan*. Additionally, *B. altitudinis*, *B. safensis* subsp. *Safensis*, and *B. paralicheniformis* were only isolated from *grep chhurpi*, *peruñyaan*, and *peha*, respectively. Chi-squared test showed a significant difference between the observed and the expected frequencies of *Bacillus* species in different fermented soybean products (Table 1). Alpha diversity analysis showed the highest number of individuals isolated from *peron namsing* followed by *peruñyaan*, *peha*, and *grep chhurpi*. However, the highest number of species were detected in *peruñyaan* samples. Similarly, highest species diversity, as per Shannon's and Simpson's diversity indices, was observed in *peruñyaan* samples and the least from *peron namsing* samples (Table 1).

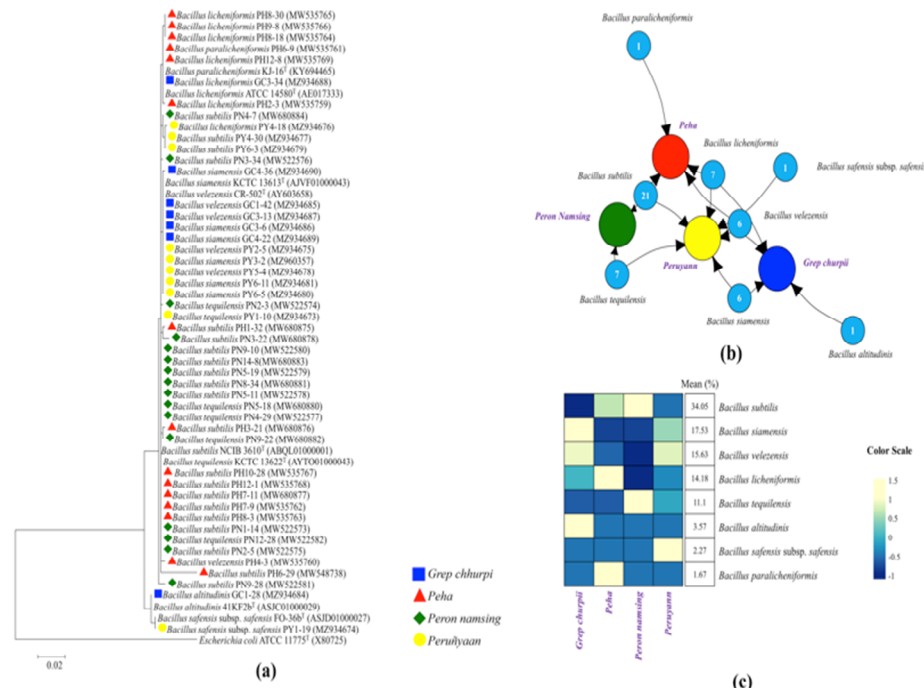

**Figure 4.** (**a**) Evolutionary relationships of taxa from *grep chhurpi*, *paeha/peha*, *peron namsing*, and *peruñyaan;* the evolutionary history was inferred using the maximum-likelihood method. (**b**) Graphical network representation of the shared and unique diverse *Bacillus* species isolated from different fermented soybean products of Arunachal Pradesh, India: *grep chhurpi*, *peha/paeha*, *peron namsing*, and *peruñyaan*. The numbers in the circles indicate the number of each species identified by 16S rRNA gene sequencing. (**c**) Distribution (%) of *Bacillus* spp., in each product.

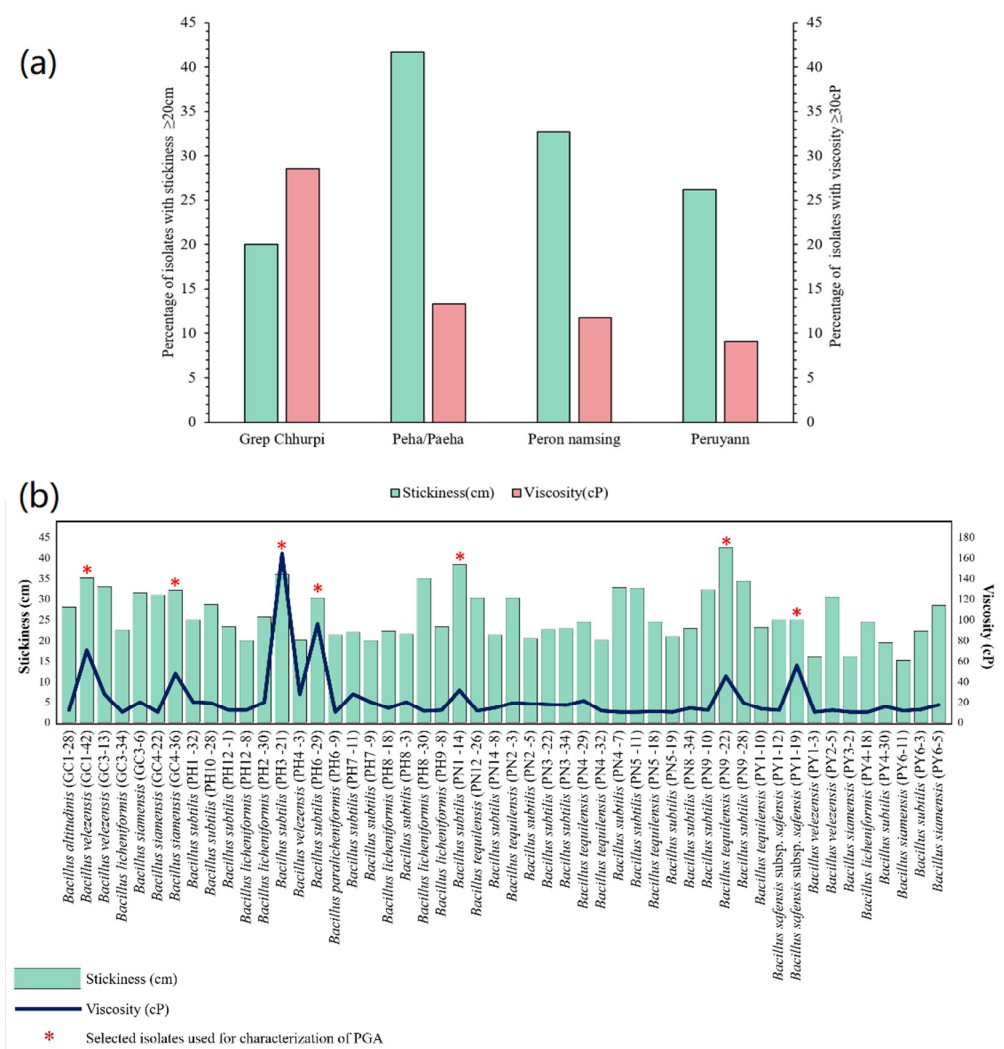

**Figure 5.** (**a**) Distribution (%) of the bacterial strains isolated from naturally fermented soybean products of Arunachal Pradesh based on stickiness and viscosity. (**b**) Graphical representation of the measurements of stickiness and dynamic viscosity of the identified *Bacillus* species isolated from fermented soybean products of Arunachal Pradesh, India.

**Table 1.** Bacterial frequency and species diversity indices of *Bacillus* species isolated from fermented soybean products of Arunachal Pradesh, India.

| Species (Strains Number) | Products (%) | | | | |
|---|---|---|---|---|---|
| | *Grep chhurpi* | *Peha* | *Peron namsing* | *Peruñyaan* | **Mean** |
| Bacillus subtilis (PH1-32, PH10-28, PH12-1, PH3-21, PH6-29, PH7-11, PH7-9, PH8-3, PN1-14, PN14-8, PN2-5, PN3-22, PN3-34, PN4-7, PN5-11, PN5-19, PN8-34, PN9-10, PN9-28, PY4-30 and PY6-3) | 0 | 53.33 | 64.71 | 18.18 | 34.05 |
| *Bacillus siamensis* (GC3-6, GC4-22, GC4-36, PY3-2, PY6-11 and PY6-5) | 42.86 | 0.00 | 0.00 | 27.27 | 17.53 |
| *Bacillus velezensis* (GC1-42, GC3-13, PH4 -3, PY1-3, PY2-5 and PY5-4) | 28.57 | 6.67 | 0.00 | 27.27 | 15.63 |

**Table 1.** *Cont.*

| Species (Strains Number) | Products (%) | | | | |
|---|---|---|---|---|---|
| | *Grep chhurpi* | *Peha* | *Peron namsing* | *Peruñyaan* | **Mean** |
| *Bacillus licheniformis* (GC3-34, PH12-8, PH2-30, PH8-18, PH8-30, PH9-8 and PY4-18) | 14.29 | 33.33 | 0.00 | 9.09 | 14.18 |
| *Bacillus tequilensis* (PN12-26, PN2-3, PN4-29, PN4-32, PN5-18, PN9-22 and PY1-10) | 0.00 | 0.00 | 35.29 | 9.09 | 11.1 |
| *Bacillus altitudinis* (GC1-28) | 14.28 | 0.00 | 0.00 | 0 | 3.57 |
| *Bacillus safensis* subsp. *Safensis* (PY1-19) | 0.00 | 0.00 | 0.00 | 9.10 | 2.27 |
| *Bacillus paralicheniformis* (PH6-9) | 0 | 6.67 | 0.00 | 0 | 1.67 |
| **Diversity indices** | | | | | |
| Taxa_S | 4.0 | 4.0 | 2.0 | 6.0 | |
| Individuals | 7.0 | 15.0 | 17.0 | 11.0 | |
| Simpson_1-D | 0.693 | 0.595 | 0.456 | 0.793 | |
| Shannon_H | 1.277 | 1.063 | 0.649 | 1.673 | |
| **Samples** | **Chi-Square, $\chi^2$ (p-value)** | | | | |
| *Peron namsing* vs *Peha* | 0.009 * | | | | |
| *Peron namsing* vs *Grep churpii* | 0.0002 * | | | | |
| *Peron namsing* vs *Peruyann* | 0.003 * | | | | |
| *Peha* vs *Grep churpii* | 0.01 * | | | | |
| *Peha* vs *Peruyann* | 0.04 * | | | | |
| *Grep churpii* vs *Peruyann* | 0.6 [NS] | | | | |

Note: * indicates significant, "NS" indicates not significant.

### 3.2. Characterization of PGA

Viscosity of 50 selected *Bacillus* spp., on the basis of high stickiness (≥20 cm), was measured, and seven *Bacillus* spp. showing ≥ 0.03Pa.s (30 cP) of viscosity were selected viz. *Bacillus velezensis* GC1-42, *B. siamensis* GC4-36, *B. subtilis* PH3-21, *B. subtilis* PN1-14, *B. subtilis* PH6-29, *B. tequilensis* PN9-22, and *B. safensis* subsp. *safensis* PY1-19 for PGA production. *B. subtilis* PH3-21 showed the highest viscosity of 0. 164Pa.s (164 cP) among the seven strains of *Bacillus* (Figure 4b). Amino acid analysis of PGA extracted from seven *Bacillus* spp. also showed the predominance of glutamic acid (60.2–78.9%) in comparison to the standard PGA (100%), with *B. safensis* subsp. *safensis* PY1-19 showing the highest percentage of glutamic acid of 78.9% (Table 2). A number of other amino acids was also detected, which could be due to contamination of secreted bacterial proteins or residual biomass remaining in the sample after centrifugation. Gel filtration chromatography showed prominent peaks in all seven PGA-producing *Bacillus* between 5–6 min (Figure 6), which corresponded to the elution time of thyroglobulin (MW = 660 kDa) and/or standard PGA, suggesting high molecular weight of PGA isolated from *Bacillus* spp.

**Table 2.** Amino acid contents of PGA extracted from *Bacillus* spp. of naturally fermented soybean foods of Arunachal Pradesh, India.

| Amino acid (pmol) | *B. velezensis* GC1-42 | *B. siamensis* GC4-36 | *B. subtilis* PH3-21 | *B. subtilis* PN1-14 | *B. tequilensis* PN9-22 | *B. safensis* subsp. *safensis* PY1-19 | *B. subtilis* PH6-29 | Standard PGA ** |
|---|---|---|---|---|---|---|---|---|
| Aspartic acid | 69.1 | 142.3 | 129.3 | 104.4 | 155.8 | 55.5 | 57.7 | ND * |
| Threonine | 80.1 | 49.0 | 52.8 | 41.8 | 49.6 | 18.7 | 13.7 | ND |
| Serine | 147.0 | 94.6 | 118.3 | 96.0 | 110.8 | 37.7 | 37.7 | ND |
| Glutamic acid | 2298.8 | 3941.8 | 4766.5 | 2608.8 | 3338.1 | 1935.8 | 1100.2 | 69.6 |
| Proline | ND | ND | ND | ND | ND | ND | ND | ND |
| Glycine | 151.3 | 364.2 | 285.9 | 212.6 | 250.5 | 89.0 | 52 | ND |
| Alanine | 189.3 | 398.1 | 469.8 | 266.7 | 493.6 | 137.3 | 492 | ND |
| Cysteine | ND | ND | ND | ND | ND | ND | ND | ND |
| Valine | ND | ND | ND | ND | ND | ND | ND | ND |
| Methionine | ND | ND | ND | ND | ND | ND | ND | ND |
| Isoleucine | ND | ND | ND | ND | ND | ND | ND | ND |
| Leucine | ND | ND | ND | ND | ND | ND | ND | ND |
| Tyrosine | 229.0 | 181.0 | 204.4 | 67.9 | 273.8 | 66.8 | 77.8 | ND |
| Phenylalanine | 271.9 | 230.8 | 295.2 | 82.4 | 482.4 | 46.5 | 55.5 | ND |
| Histidine | 85.7 | 55.0 | 50.8 | 33.3 | 67.5 | 18.3 | 67.2 | ND |
| Lysine | 232.2 | 89.5 | 123.6 | 45.7 | 95.8 | 39.0 | 95.2 | ND |
| Arginine | 65.5 | 36.9 | 45.5 | 48.3 | 14.1 | 9.3 | 15.1 | ND |
| Total | 3819.9 | 5583.2 | 6542.1 | 3607.8 | 5332.0 | 2453.8 | 2064 | 69.6 |
| Glutamic acid (%) | 60.2 | 70.6 | 72.9 | 72.3 | 62.6 | 78.9 | 53.3 | 100.0 |

** purified commercial standard PGA (Sigma-Aldrich) used as a reference; * ND; not detected.

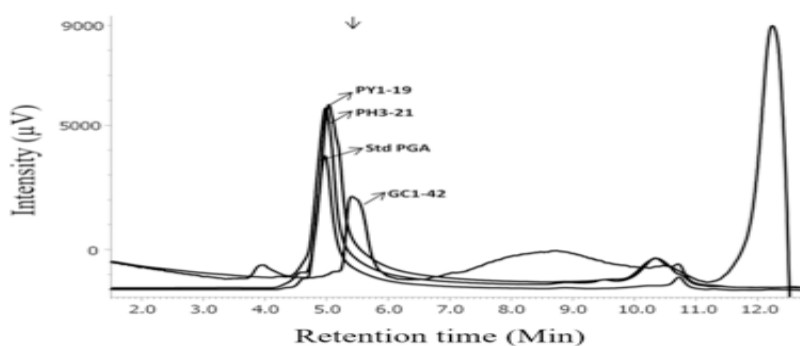

**Figure 6.** The relative molecular weight was determined in relation to elution of standard markers (indicated in arrows), thyroglobulin (mw = 660 kDa). The chromatograms of *B. safensis* subsp. *safensis* PY1-19 shows the highest peak of 78.9% of glutamic acid, followed by *B. subtilis* PH3-21 (72.3%) and the lowest by *B. velezensis* GC1-42 (60.2%) along with standard PGA. The elution time for all other samples remained similar.

### 3.3. Selection of High-PGA-Producing B. safensis subsp. safensis PY1-19

*B. safensis* subsp. *safensis* PY1-19, isolated from *peruñyaan*, with its high glutamic acid of 78.9% showed no hemolytic activity but showed ability to produce phytase (data not shown). Additionally, it also showed antagonistic properties against *Escherichia coli* MCC

2413, *Salmonella enterica* subsp. *enterica* MTCC 3223 and *Staphylococcus aureus* MTCC 740 (data not shown). *Peruñyaan* produced by using pure strain of *B. safensis* subsp. *safensis* PY1-19 showed high stickiness and viscosity and produced PGA of around 11 g/L (Figure 7).

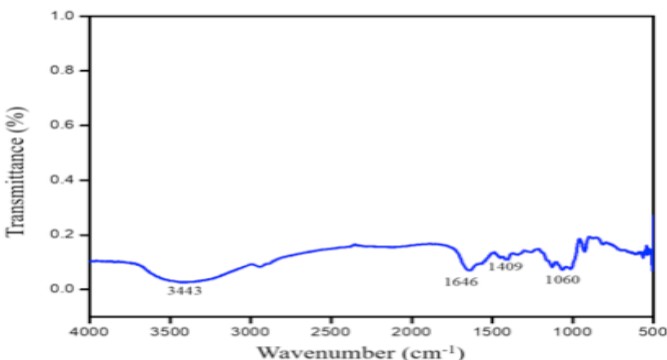

**Figure 7.** FTIR absorption spectrum of γ-PGA produced from selective fermentation by *B. safensis* subsp. *safensis* PY1-19 from *peruñyaan*.

## 4. Discussion

Use of beneficial microorganisms for food fermentation is common practice in India among different ethnic communities [34]. The ethnic communities of Arunachal Pradesh in India have historically and culturally adopted the consumption of sticky fermented soybean foods in their food habit, which is uncommon in many regions of India except in northeastern regions of India. Due to multi-ethnicity, several sticky and *umami* [5] flavored fermented soybean foods are preferred by every ethnic community in Arunachal Pradesh, and some of them are common viz. *grep chhurpi, peha/paeha, peron namsing,* and *peruñyaan*. Documenting the preparation is also a necessity for preserving traditional knowledge, which is dying a slow death with modernization replacing it. Since naturally fermented soybean food is a hub for endospore forming, Gram-positive, non-pathogenic, and beneficial *Bacillus* spp. [19,35,36], which are also a natural source of PGA production, we screened the PGA-producing *Bacillus* from *grep chhurpi, peha/paeha, peron namsing*, and *peruñyaan*. Since stickiness is one of the preferred quality criteria for fermented soybean foods by the consumers [7], we first screened the isolates for their stickiness, and on the basis of high stickiness (>20 cm), 50 bacterial isolates were selected and identified up to species level. High-PGA-rich fermented soybean is highly mucilaginous with health benefits.

Interspecies diversity of *Bacillus* was observed in samples, which displayed a dominance of *Bacillus subtilis*, followed by *B. siamensis, B. velezensis, B. licheniformis, B. tequilensis, B. altitudinis, B. safensis* subsp. *Safensis,* and *B. paralicheniformis.* The bacterial diversity is dependent on a number of factors, such as place of fermentation, altitude of the place, length of the fermentation, temperature of fermentation, and substrate used [37]. *B. subtilis* is the most predominant bacterium in many Asian fermented soybean foods with multiple functionalities [10,11,19,27,37,38]. *B. siamensis*, a bacterium with potent anti-fungal activity [39], has been reported from *doenjang*, fermented soybean food of Korea [40]. *B. velezensis*, a functional bacterium, is reported from *douchi*, fermented soybean food of China [41]. *B. licheniformis* was reported from *kinema, tungrymbai* and *bekang,* fermented soybean foods of North East India [32,33]. Biosurfactant-producing *B. tequilensis* was isolated from *kinema* [42]. *B. altitudinis* is an oligotrophic organism and plant-growth promoter [43], which was isolated from samples of *grep chhurpi* and *peruñyaan*. *B. paralicheniformis* was reported from *cheonggukjang* of Korea [11]. B. *safensis* was reported from some fermented soybean foods of Korea [44] and also from *ugba*, a fermented African oil beans [45]. However, this is the first report of B. *safensis* subsp. *safensis* producing PGA from Indian fermented soybean that also aids in PGA production. Chi-squared test showed a significant difference between the observed and the expected frequencies of *Bacillus*, which indicated a significant diversity of *Bacillus* species among the products.

Though all 50 isolates were potential candidates for PGA production, however, only 7 isolates stood out in production of high viscous PGA, which had a high molecular weight of ≥660 kDa. Molecular weight and viscosity are linked with one another since viscosity is an important factor for good PGA production [46,47]. It was observed in the present study that glutamic acid content was as high as 79%. High viscosity of media is a good characteristic of the organisms that are capable of producing polyglutamic acid in the fermented soybean foods [23]. The γ-PGA is naturally produced by *Bacillus* spp., in many fermented soybean foods, such as Japanese *natto* [9], Korean *cheonggokjang* [15] and Himalayan *kinema* [8]. PGA with high molecular weight has a number of uses in waste management and the food industry [23] and is also used as a viscosity enhancing agent usually in dye and heavy metal removal as well as a flocculating agent [13].

*B. safensis* subsp. *safensis* PY1-19, isolated from *peruñyaan*, showed 78.9% of glutamic acid, an ability to produce phytase, and antagonistic properties against pathogenic bacteria with no hemolytic activity. Hence, this strain was selected as a potential starter to prepare *peruñyaan* at laboratory and also was examined for PGA production. The final product showed high stickiness and viscosity and produced PGA of around 11g/L. *B. safensis* subsp. *safensis* also showed phytase production, which reduces phytic acid, a major anti-nutritive factor in soybean [48]. Non-hemolytic and antagonistic properties of *B. safensis* subsp. *safensis* PY1-19 may project it as a safe starter culture with a high PGA producer. Though a number of *Bacillus* spp. has been given a qualified presumption of safety (QPS) by the European Food Safety Authority (EFSA) [49] and also listed in microbial food culture (MFC) inventory [50], *Bacillus* spp. in foods have to be properly studied for their harmful properties as well [27]. Though, *B. safensis* subsp. *safensis* has been utilized for a number of uses, such as phytoremediation of oil contaminated saline soils [51]; antifungal activity [52], however, and its activity in PGA production have also not been studied so far. High-PGA-rich foods have been associated with an anti-obesity effect, lowering blood glucose, and reducing of cardiovascular diseases [15]. Optimization of the PGA production as in the cases of *Bacillus siamensis* SB1001 [53] and *Bacillus subtilis* NX-2 [54] could be boosted for high production of PGA from selected isolates for further work.

## 5. Conclusions

This study focuses on diversity of *Bacillus* spp., which are capable of producing PGA from naturally fermented soybean foods of Arunachal Pradesh in India. Species of *Bacillus*, isolated from fermented soybean foods, are not only capable of producing high stickiness in fermented soybean but they also produce PGA with high molecular weight. The capability of functional *B. safensis* subsp. *safensis* PY1-19, isolated from *peruñyaan,* to produce high viscous PGA may be further explored to develop a potential starter culture for production of optimized and desirable fermented soybean foods. Future studies may be initiated to enhance, extract, and purify the inexpensive PGA from natural food sources for industrial applications as well for food supplements.

**Author Contributions:** Conceptualization, J.P.T. and P.R.; methodology, P.P. and P.S.Y.; software, P.P. and P.S.Y.; validation, J.P.T. and P.R.; formal analysis, P.P and P.S.Y., investigation, P.P. and P.S.Y.; resources, J.P.T.; data curation, J.P.T.; writing—original draft preparation, P.P.; writing—review and editing, J.P.T.; visualization, P.P. and P.S.Y.; supervision, J.P.T. and P.R.; project administration, J.P.T.; funding acquisition, J.P.T. All authors have read and agreed to the published version of the manuscript.

**Funding:** The authors are grateful to the Department of Biotechnology (DBT), Ministry of Science and Technology, Government of India for financial support through DBT project, sanction number:BT/PR24616/NER/95/778/2017.

**Institutional Review Board Statement:** Not applicable.

**Informed Consent Statement:** Not applicable.

**Data Availability Statement:** All identified 16S rRNA gene sequences were deposited in GenBank-NCBI under the accession numbers: MW522573-MW522582; MW535759-MW535769; MW680875-MW680884, MW548738, MZ934672- MZ934681, MZ960357, and MZ934684-MZ934690.

**Acknowledgments:** Authors are grateful to Department of Biotechnology (DBT), Govt. of India for financial support. P.P. is grateful to DBT for JRF and SRF in the Research project sanctioned to JPT.

**Conflicts of Interest:** The authors declare to have no conflict of interests.

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
