# Peer review of "Screening of Poly-Glutamic Acid (PGA)-Producing Bacillus Species from Indian Fermented Soybean Foods and Characterization of PGA"

_fermentation, doi:10.3390/fermentation8100495_

Round 1

Reviewer 1 Report

The paper under consideration deals with the identification of microbial agents of soya fermented traditional products and the selection of species for preparation of starters based on main properties of those products as viscosity and stickiness from PGA produced by Bacillus spp.

This is a well-structured and detailed article concerning the introduction, methods and result presentation, obtained from a good experimental plan, despite the conclusions being very short in relation to the results presented which should be reviewed.

These are the most relevant generic comments about the submitted paper and I have no further detailed suggestions as the is well written, although english is not my natural language.

In addition to the revision of the conclusions, my only suggestion concerns the unit in which viscosity is expressed, Cp, questioning whether it will not be possible to replace it with the SI unit of expression, Pa.s.

Author Response

Reviewer # 1

The paper under consideration deals with the identification of microbial agents of soya fermented traditional products and the selection of species for preparation of starters based on main properties of those products as viscosity and stickiness from PGA produced by Bacillus spp. This is a well-structured and detailed article concerning the introduction, methods and result presentation, obtained from a good experimental plan, despite the conclusions being very short in relation to the results presented which should be reviewed.

These are the most relevant generic comments about the submitted paper and I have no further detailed suggestions as the is well written, although English is not my natural language.

In addition to the revision of the conclusions, my only suggestion concerns the unit in which viscosity is expressed, Cp, questioning whether it will not be possible to replace it with the SI unit of expression, Pa.s.

Answer: Centipoise (cP) was used in the text since the Viscometer used gave us reading in cP but it is possible to change the unit of the expression to Pa.s. Hence as suggested by the Reviewer,  changed to Pa.s. in the revised manuscript (1000cP=1Pa.s.)

Reviewer 2 Report

The manuscript ‘Screening of Poly-glutamic acid (PGA)-producing Bacillus species from Indian fermented soybean foods and characterization of PGA’ is an intriguing work that discusses the isolation of Bacillus sp from fermented foods that produce polyglutamate and also describes the characterization of polyglutamate. This manuscript is recommended for publication in the fermentation journal after minor revision.

Minor comments:

Do you mean other parts of India in the introduction section?

Font size in fig. 1 can be improved for a better understanding

Include one or two health benefits or uses of polyglutamate in the introduction section as well

37oC for 30 min. Correct the symbol

Fig. 2 figure resolution can be improved

Why is fig. 2 in the methods section instead of the results section?

Gram-positive. Why caps letters in the middle of a sentence?

On page 12, discussion section the bacterial species are not italicized. Check throughout the manuscript thoroughly

Check grammar, and typos in the manuscript

Author Response

Reviewer # 2

The manuscript ‘Screening of Poly-glutamic acid (PGA)-producing Bacillus species from Indian fermented soybean foods and characterization of PGA’ is an intriguing work that discusses the isolation of Bacillus sp from fermented foods that produce polyglutamate and also describes the characterization of polyglutamate. This manuscript is recommended for publication in the fermentation journal after minor revision.

Minor comments:

Q 1: Do you mean other parts of India in the introduction section?

Answer: Due to a typographical error the sentence was wrong and we have corrected it and changed it from “their parts of India” to “other parts of India” in the revised introduction. 

Q 2: Font size in fig. 1 can be improved for a better understanding

Answer: As suggested by the Reviewer, we changed the font of the writings in figure 1 to improve its accessibility in the revised manuscript.

Q 3: Include one or two health benefits or uses of polyglutamate in the introduction section as well 37oC for 30 min. Correct the symbol

Answer: As suggested by the Reviewer, we included the health benefits credited to PGA in the revised introduction. We also changed the symbol for “37oC for 30 min” to “37°C for 30 min.

Q 4: Fig. 2 figure resolution can be improved

Answer: As suggested by the Reviewer, the resolution for figure 2 was also enhanced for better accessibility as well.

Q 5: Why is fig. 2 in the methods section instead of the results section?

Answer: As suggested by the Reviewer, the figures have been rearranged in the revised manuscript and in doing so Figure 2 (now Figure 4) is placed in the Results section in the revised introduction.

Q 6: Gram-positive. Why caps letters in the middle of a sentence?

Answer: This was owing to a typographical error and has been corrected in the text.

Q 7: On page 12, discussion section the bacterial species are not italicized. Check throughout the manuscript thoroughly

Answer: As suggested by the Reviewer, all the scientific were italicized in the revised manuscript.

Q 8: Check grammar, and typos in the manuscript

Answer: As suggested by the Reviewer, we have thoroughly checked the grammar and typos in the revised manuscript.

#Note: All the images have been rearranged owing which their figure numbers have changed and have all been placed in appropriate places as per requirement. All the resolution of the images has also been enhanced. 
